# Modeling Category-Selective Cortical Regions with Topographic Variational Autoencoders

**T. Anderson Keller**\*
UvA-Bosch Delta Lab
University of Amsterdam

**Qinghe Gao**\*
ChemEngAI Lab
Delft University of Technology

**Max Welling**
UvA-Bosch Delta Lab
University of Amsterdam

## Abstract

Category-selectivity in the brain describes the observation that certain spatially localized areas of the cerebral cortex tend to respond robustly and selectively to stimuli from specific limited categories. One of the most well known examples of category-selectivity is the Fusiform Face Area (FFA), an area of the inferior temporal cortex in primates which responds preferentially to images of faces when compared with objects or other generic stimuli. In this work, we leverage the newly introduced Topographic Variational Autoencoder to model of the emergence of such localized category-selectivity in an unsupervised manner. Experimentally, we demonstrate our model yields spatially dense neural clusters selective to faces, bodies, and places through visualized maps of Cohen's d metric. We compare our model with related supervised approaches, namely the Topographic Deep Articifial Neural Network (TDANN) of Lee et al. [39], and discuss both theoretical and empirical similarities. Finally, we show preliminary results suggesting that our model yields a nested spatial hierarchy of increasingly abstract categories, analogous to observations from the human ventral temporal cortex.

## 1 Introduction

Category-selectivity is observed throughout the cerebral cortex. At a high level it describes the observation that certain localized regions of the cortical surface have been measured to respond preferentially to specific stimuli when compared with a set of alternative control images. It has been measured across a diversity of species [29, 57, 44], directly through fMRI and neural recordings [50], and more indirectly through observational studies of patients with localized cortical damage [43]. Examples of category-selective areas in the visual stream include the Fuisform Face Area (FFA) [29], the Parahippocampal Place Area (PPA) [17, 44], and the Extrastriate Body Area (EBA) [48] which respond selectively to faces, places, and bodies respectively. However, the extent of category-selectivity does not stop at such basic categories. Instead, selective maps have been observed for both more abstract 'superordinate' categories, such as animacy versus inanimacy [22, 35], as well as for more fine-grained 'subordinate' categories such as human-faces versus animal-faces [21]. These maps are seen to be superimposed on one-another such that the same cortical region expresses selectivity simultaneously to animate objects and human-faces, while other spatially disjoint regions are simultaneously selective to inanimacy and 'places' (images of scenes). Such overlapping maps have been interpreted by some researchers as nested hierarchies of increasingly abstract categories, potentially serving to increase the speed and efficiency of classification [19].

In interpreting these observations, one may naturally wonder as to the origins of such localized specialization. From the current literature, the driving factors can roughly be divided into two potentially complimentary categories: anatomical, and information theoretic. Anatomically, the

---

\*Equal contribution. Work done while Qinghe Gao was interning at UvA-Bosch Delta Lab.

3rd Workshop on Shared Visual Representations in Human and Machine Intelligence (SVRHM 2021) of the Neural Information Processing Systems (NeurIPS) conference, Virtual.

arrangement and properties of different cell bodies can be observed to vary slightly in different regions of the cortex in loose alignment with category selectivity [59, 12, 55], possibly serving as an innate blueprint for specialization. In the same category, the principle of 'wiring length minimization' [36, 18] posits that evolutionary pressure has encouraged the brain to reduce the cumulative length of neural connections in order to reduce the costs associated with the volume, building, maintenance, and use of such connections. Computational models which attempt to integrate such wiring length constraints [39, 63, 10] have recently have been observed to yield localized category selectivity such as 'face patches' similar to those of macaque monkeys. A hypothesized second factor behind the emergence of category specialization, which has recently gained increasing empirical support, derives its explanatory power from information theory. Empirical studies have discovered that sufficiently deep convolutional neural networks naturally learn distinct and largely separate sets of features for certain domains such as faces and objects. Specifically, the work of Dobs et al. [16], showed that feature maps in the later layers of deep convolutional neural networks can be effectively segregated into object and face features such that lesioning one set of feature maps does not significantly impact performance of the network on classification of the other data domain. Such experiments, and others [5, 33], suggest that the specialization of neurons may simply be an optimal code for representing the natural statistics of the underlying data when given a sufficiently powerful feature extractor.

Pursuant to these ideas, this work proposes that a single underlying information theoretic principle, namely the principle of redundancy reduction [7], may account for localized category selectivity while simultaneously serving as a principled unsupervised learning algorithm. Simply, the principle of redundancy reduction states that an optimal coding scheme is one which minimizes the transmission of redundant information. Applied to neural systems, this describes the ideal network as one which has statistically maximally independant activations – yielding a form of specialization. This idea served as the impetus for computational frameworks such as Sparse Coding [45] and Independant Component Analysis (ICA) [8, 15, 24, 25]. Interestingly, however, further work showed that features learned by linear ICA models were not entirely independant, but indeed contained correlation of higher order statistics. In response, researchers proposed a more efficient code could be achieved by modeling these residual dependencies with a hierarchical topographic extension to ICA [26, 28], separating out the higher order 'variance generating' variables, and combining them locally to form topographically organized latent variables. Such a framework shares a striking resemblance to models of divisive normalization [41, 6], but inversely formulated as a generative model. Ultimately, the features learned by such models were reminiscent of pinwheel structures observed in V1, encouraging multiple comparisons with topographic organization in the biological visual system [27, 28, 42].

In this work, we leverage the recently introduced Topographic Variational Autoencoder [31, 30], a modern instantiation of such a topographic generative model, and demonstrate that it is capable of modeling localized category selectivity as well as higher order abstract organization, guided by a single unsupervised learning principle. We quantitatively validate category selectivity through visualization of Cohen's d effect size metric for different image classes, showing selective clusters for faces, bodies, and places. We compare our model with another recently developed model of topographic organization, the Topographic Deep Artificial Neural Network (TDANN) [39], and demonstrate qualitatively similar results with an unsupervised learning rule. Finally, we show preliminary results indicating that our model contains a nested spatial hierarchy of increasingly abstract categories, similar to those observed in the human ventral temporal cortex [21, 19].

## 2   Related Work

Recently, a number of models of topographic organization in the visual system have been developed leveraging modern deep neural networks. Zhang et al. [63] demonstrated category-selective regions, as well as a nested spatial hierarchy of selectivity, through the use of self-organizing maps (SOMs). Due to the challenges with scaling SOMs, the inputs were dimensionality-reduced with PCA, limiting the applicability of the algorithm to arbitrary neural network architectures. Concurrently with our work, Blauch et al. [10] developed the Interactive Topographic Network (ITN), inducing local correlation through locally-biased excitatory feedforward connections in a biologically-constrained model. Most related to our work, the TDANN of Lee et al. [39] incorporated a biologically derived proxy for wiring length cost into the fully connected layers of a supervised Alexnet model [37], and similarly demonstrated emergent localized category-selectivity. Our model differs from these in that it explicitly formulates a properly normalized density over the input data with topographic

organization originating as a prior over latent the variables – thereby unifying feature extraction and topographic organization into a single training objective: maximization of the data likelihood.

# 3 Background

**Variational Autoencoders**  Bayesian modeling, the theoretical framework underlying probabilistic generative models, has been proposed in multiple studies as a potential model of human learning [1, 56]. Abstractly, the goal of a generative model is to accurately capture the true data generating process. Latent variable models propose to achieve this by defining a joint distribution over observations $\mathbf{X}$ and unobserved latent variables $\mathbf{T}$, commonly assuming that the joint factorizes into the product of a conditional 'generative' distribution and a prior: $p_{\mathbf{X},\mathbf{T}}(\mathbf{x},\mathbf{t}) = p_{\mathbf{X}|\mathbf{T}}(\mathbf{x}|\mathbf{t})p_{\mathbf{T}}(\mathbf{t})$. These distributions are often parameterized with deep neural networks, earning the title 'Deep Latent Variable Models'. The goal of training such generative models is then to maximize the marginal likelihood of the data, $p_\theta(\mathbf{x}) = \int p_\theta(\mathbf{x},\mathbf{t})d\mathbf{t}$, with respect to the parameters $\theta$. However, due to the intractability of the integral and the true posterior $p_\theta(\mathbf{t}|\mathbf{x})$, this is almost always intractable to compute exactly. Approximate solutions such as the Variational Autoencoder (VAE) [32] were thus developed to provide tractable lower bounds on the data likelihood. Simply, in the VAE framework, an approximate posterior $q_\phi(\mathbf{t}|\mathbf{x})$ for the latent variables is separately parameterized and optimized to be close to the true posterior through the Evidence Lower Bound (ELBO):

$$\mathcal{L}_{\theta,\phi}(\mathbf{x}) = \mathbb{E}_{q_\phi(\mathbf{t}|\mathbf{x})}\left(\log p_\theta(\mathbf{x}|\mathbf{t}) - D_{KL}[q_\phi(\mathbf{t}|\mathbf{x})||p_{\mathbf{T}}(\mathbf{t})]\right) \leq \log p_\theta(\mathbf{x})$$

Through use of the reparameterization trick [32, 52], all parameters $\phi$ and $\theta$ can then be simultaneously optimized with stochastic gradient descent.

**Topographic Generative Models**  In standard generative models such as ICA or Variational Autoencoders (VAEs) [52, 32], it is common to define a prior over latent variables such that all variables are independant. Topographic generative models differ from this by instead having a more complex correlation structure defined by the spatial distance between variables in a pre-defined topographic layout. In Topographic ICA (TICA) [26], such a local-correlation structure was shown to be efficiently achievable through a 2-layer hierarchical generative model. Specifically, at the highest layer, a set of 'variance generating' variables $\mathbf{V}$ are independently sampled and subsequently summed in local neighborhoods to determine the variance of lower level topographic variables $\mathbf{T}$. Formally, for independant normal variables $\mathbf{V} \sim \mathcal{N}(\mathbf{0}, \mathbf{I})$, the variances of $\mathbf{T}$ are given by a non-linearity $\phi$ applied to local sums of $\mathbf{V}$: $\boldsymbol{\sigma} = \phi(\mathbf{W}\mathbf{V})$, where we have expressed the local-sum operation in matrix form here for conciseness. A well known example of such a 'local-sum' matrix $\mathbf{W}$ is the matrix representation of the convolution operation. The vector $\mathbf{T} \sim \mathcal{N}(\mathbf{0}, \boldsymbol{\sigma}^2\mathbf{I})$ can then be seen to have correlations of variance across elements $T_i$ & $T_j$ if these elements are 'connected by $\mathbf{W}$', and thus share a subset of their variance generating variables $V$. Interestingly, the proposed wiring length proxy employed in the TDANN [39] turns out to be based on the same underlying statistical property as the topographic generative models described in this work, namely local correlation. This suggests that these ideas may not be mutually exclusive, and hints at a potential fundamental connection between wiring length minimization and a generative modeling perspective of the brain.

**The Topographic VAE**  Inspired by linear topographic generative models such as TICA, the Topographic Variational Autoencoder (TVAE) [31] was recently introduced to train deep *nonlinear* latent variable models with topographic structure. The model places a Topographic Product of Student's-T prior [60, 46] over the latent variables, and achieves efficient training through a hierarchical construction identical to that of TICA. Formally, the model parameterizes the conditional generative distribution with a powerful function approximator $p_\theta(\mathbf{x}|g_\theta(\mathbf{t}))$, and trains the paramters of this model through the use of *two* approximate posteriors $q_\phi(\mathbf{z}|\mathbf{x})$ and $q_\gamma(\mathbf{u}|\mathbf{x})$ which are combined to construct the topographic $\mathbf{t}$ variable. Explicitly:

$$q_\phi(\mathbf{z}|\mathbf{x}) = \mathcal{N}\big(\mathbf{z}; \mu_\phi(\mathbf{x}), \sigma_\phi(\mathbf{x})\mathbf{I}\big) \qquad q_\gamma(\mathbf{u}|\mathbf{x}) = \mathcal{N}\big(\mathbf{u}; \mu_\gamma(\mathbf{x}), \sigma_\gamma(\mathbf{x})\mathbf{I}\big) \tag{1}$$

$$\mathbf{t} = \frac{\mathbf{z} - \mu}{\sqrt{\mathbf{W}\mathbf{u}}} \qquad p_\theta(\mathbf{x}|g_\theta(\mathbf{t})) = p_\theta(\mathbf{x}|g_\theta(\mathbf{z}, \mathbf{u})) \tag{2}$$

The parameters $\theta, \phi, \gamma$ and $\mu$ are then optimized to maximize the likelihood of the data through the Evidence Lower Bound (ELBO):

$$\mathbb{E}_{q_\phi(\mathbf{z}|\mathbf{x})q_\gamma(\mathbf{u}|\mathbf{x})}\left(\log p_\theta(\mathbf{x}|g_\theta(\mathbf{t})) - D_{KL}[q_\phi(\mathbf{z}|\mathbf{x})||p_{\mathbf{Z}}(\mathbf{z})] - D_{KL}[q_\gamma(\mathbf{u}|\mathbf{x})||p_{\mathbf{U}}(\mathbf{u})]\right) \tag{3}$$

## 4 Methods

**Evaluation**    Following prior computational work [39, 63] and fMRI studies [4], we use Cohen's $d$ metric [14, 54], a measure of standardized difference of two means, as our selectivity metric. Given the means $\bar{m}_1$ & $\bar{m}_2$ and standard deviations $\sigma_1$ & $\sigma_2$ of two sets of data, the $d$ metric is given as:

$$d = \frac{\bar{m}_1 - \bar{m}_2}{\sqrt{\frac{1}{2}\left(\sigma_1^2 + \sigma_2^2\right)}} \tag{4}$$

This value is unitless and can be seen as expressing the difference between two means in terms of units of 'pooled variability'. In this work, the mean $\bar{m}_1$ corresponds to the mean activation of a single neuron computed across an entire dataset of class-specific target images (e.g. faces), while $\bar{m}_2$ is the mean activation of the same neuron across a dataset of control images which do not contain this class.

**Datasets**    The dataset used for training both the TDANN and TVAE is a composition of the ImageNet 2012 [53] and Labeled Faces in the Wild (LFW) datasets [23], following Lee et al. [39]. The TDANN was trained to classify the 1000 distinct image classes from ImageNet, plus one generic face class encompassing all of LFW. The TVAE used no such class labels. To measure the category selectivity of the models, the primary test face dataset used in Figures 1, 2, & 4 was a ∼25,000 image subset of VGGface2 [11]. The control 'object' dataset for Figures 1 & 4 was composed of 25,000 images from the validation set of ImageNet. To measure selectivity to body parts and places in Figure 2, we created a 'body' dataset composed of headless body images [61] and hands [3], and used the Place365 dataset [65] for places. In Figure 2, the 'control' set used for each class was defined to be the compliment of the test set, i.e. all other datasets besides the target category of interest.

**Models**    All models are trained on top of features extracted by the final convolutional layer of a pre-trained Alexnet model [37, 47]. The Alexnet architecture was chosen to match the setup from Lee et al. [39] and Zhang et al. [63], and has further been shown to have remarkable similarities to hierarchical processing in the human visual stream [62, 13, 20]. For the TVAE, we randomly initialize and train a single linear layer encoder and decoder with 4096 output neurons, arranged in a 64x64 grid with circular boundary conditions to avoid edge effects. For the TDANN, we randomly initialize and train all three fully connected layers of Alexnet, imposing the spatial correlation loss over both 'FC6' and 'FC7'. In the following, all selectivity maps are displayed for 'FC6', following Lee et al. [39]. All hyperparameter and training details can be found in Section B.1.

## 5 Experiments

In the following, we explore the category-selectivity of top-level neurons trained with the Topographic VAE framework on realistic images. We observe that neurons do indeed become category-selective, and that selective neurons tend to group together to form localized category-selective regions for a variety of domains including faces, bodies, and places. We compare these results with a non-topographic baseline (pre-trained Alexnet), and a re-implementation of the TDANN [39], observing qualitatively similar results. Additionally, following Zhang et al. [63], we plot selectivity maps to more abstract concepts (such as animacy and real-world size), and observe that such maps overlap in an intuitive manner, suggesting the existence of a nested spatial hierarchy of categories.

**Localized Category-Selectivity**    In Figure 1, we plot the continuous value of Cohen's $d$ metric for all neurons as arranged in a 2-d grid. The baseline (left) shows the first fully connected layer (FC6) of a pre-trained Alexnet architecture. As expected, the neurons of this model have no defined spatial organization and thus result in a random selectivity map. We note the existence of class-selective neurons is not guaranteed, but their appearance here is in-line with observations from prior work [39, 51, 2]. Secondly, we compare our TVAE model (middle) with our re-implementation of the TDANN [39] (right). We observe that both models demonstrate the emergence of face-selective clusters of comparable size and density. We see that the TVAE framework appears to yield smoother topographic maps, perhaps due to the unified objective function and unsupervised learning rule when compared with the competing supervised classification loss and wiring cost regularization of the TDANN. To validate the robustness and significance of these category selective regions, in Section A of the appendix we plot selectivity maps across four different test face datasets and four random initalizations, observing qualitatively similar clusters all settings.

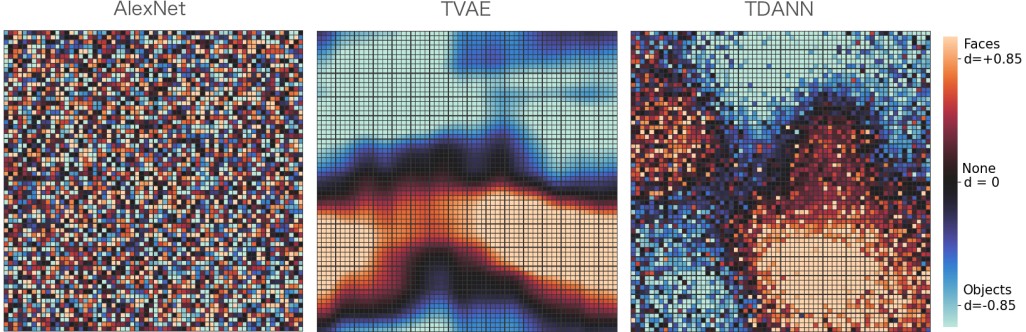

Figure 1: Face vs. Object selectivity for a non-topographic baseline, Topographic VAE, and TDANN. We see the TVAE has an emergent face cluster qualitatively similar to that of the TDANN.

**Face, Body & Place Clusters** Next, in Figure 2, we plot the simultaneous selectivity of neurons in our TVAE model to multiple classes including faces, bodies, and places. To create a map of multi-class selectivity, we follow prior work and threshold the $d$ metric at $0.85$, considered a 'strong effect' [54] and computed to be to be equivalent to a threshold of $0.65$ for noisy neural recordings in monkeys [39]. In the plot we observe an overlap of neurons with selectivity to faces and bodies, as seen in prior computational work [39] and fMRI studies [49, 58]. In Figure 5 of Section A, we see that the size and relative placement of these clusters is again consistent across multiple random initalizations.

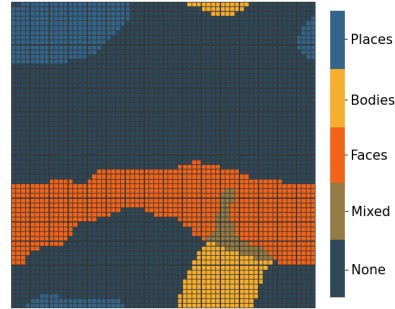

Figure 2: TVAE selectivity $d \geq 0.85$

**Impact of Topography on Model Performance** To measure the impact of the imposed topographic organization on the above models, and ensure the learned representations are not degenerate, we compare the model performance of the TDANN and TVAE with their respective non-topographic counterparts. Although the models in this study were not tuned to maximize such performance, we observe that both topographic models perform similarly to their non-topographic counterparts. Specifically the TDANN achieves 40.5% top-1 accuracy on the Imagenet validation set (+ 1 face class) versus the 45.5% top-1 accuracy of an identically trained model without spatial correlation loss. Similarly, a baseline VAE of the same architecture as the TVAE achieves roughly 3.4 bits per dimension (BPD) while the Topographic VAE achieves roughly 3.6 BPD in the same number of iterations. These results appear consistent with the intuition that topographic organization does not prevent learning, but rather acts as an inductive bias on the model, regularizing its performance. In future work we hope to quantify this regularization effect more precisely and determine in which situations it may be beneficial for generalization or computational efficiency.

**Locally Distributed Activations** To understand better how exactly individual images are represented by the TVAE, we present the activation maps corresponding to a single image from an array of classes in Figure 3. We see the representation of each image is still distributed, but most strongly activates in the associated category-selective region.

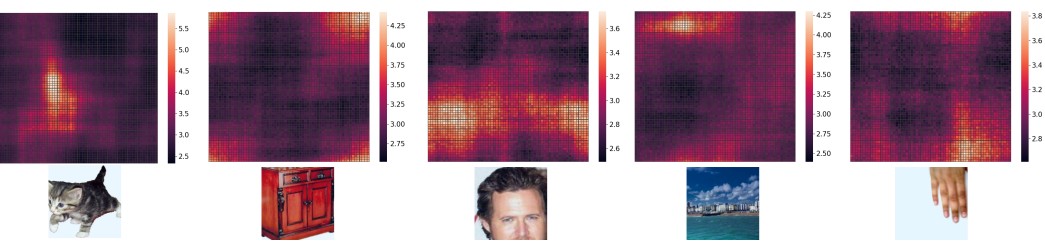

Figure 3: Activations for single images. Left to Right: Animate, Inanimate, Faces, Places, & Hands

**Nested Spatial Hierarchy of Categories**    Following Zhang et al. [63] we additionally measure the selectivity maps of our TVAE model with respect to more abstract categories such as animacy and real-world object size, obtaining such datasets from the Konkle lab database [35, 34]. Specifically, Figure 4 shows Cohen's $d$ maps (from $-1$ to $+1$) for animate versus inaminate objects (top), and for big versus small objects (middle), overlayed on the face versus object map (bottom). At the largest scale, we observe an intuitive overlap of spatial maps, specifically inanimate objects, large objects, and the place cluster from Figure 2 all overlap in the top left and right corners of the map. We additionally highlight the maximum activating neurons for three separate input images. We see the image of a red dresser activates a region which is simultaneously selective to places, large, and inanimate objects, echoing the nested spatial hierarchies thought by Grill-Spector & Weiner [19] to exist in the brain. In Section A, we again see that such a hierarchy appears consistently across four random initalizations.

# 6    Discussion

In this work we demonstrate the ability of topographic generative models, namely Topographic Variational Autoencoders, to model the emergence of category-selective cortical areas as well as more abstract spatial category hierarchies. We see the model agrees qualitatively with prior work and observations from neuroscience while being founded on a single information theoretic principle.

We note that this study is inherently preliminary and is limited by both the small size of the models used, as well as the feature extraction by a pre-trained convolutional model. It is possible that class-level features and even hierarchical organization are already partially present in some form in the 9216-dimensional feature vectors used as input, and thus it is unclear how much feature extraction the TVAE model is itself learning. Nevertheless, we highlight that there is nothing fundamentally limiting the TVAE framework from extending to train full deep convolutional networks end-to-end. This is in contrast to the existing related methods which either require a supplementary learning signal to guide feature extraction [39], or do not scale to high dimensional inputs [63].

In future work, we intend to explore hierarchical extensions of the TVAE, modeling topographic organization of features at multiple levels of the visual processing pipeline while simultaneously training directly on raw pixel inputs. Such a model would validate the idea of end-to-end unsupervised category-selectivity while simultaneously providing a learned decoder from latent space to image space, opening new avenues for experimentation.

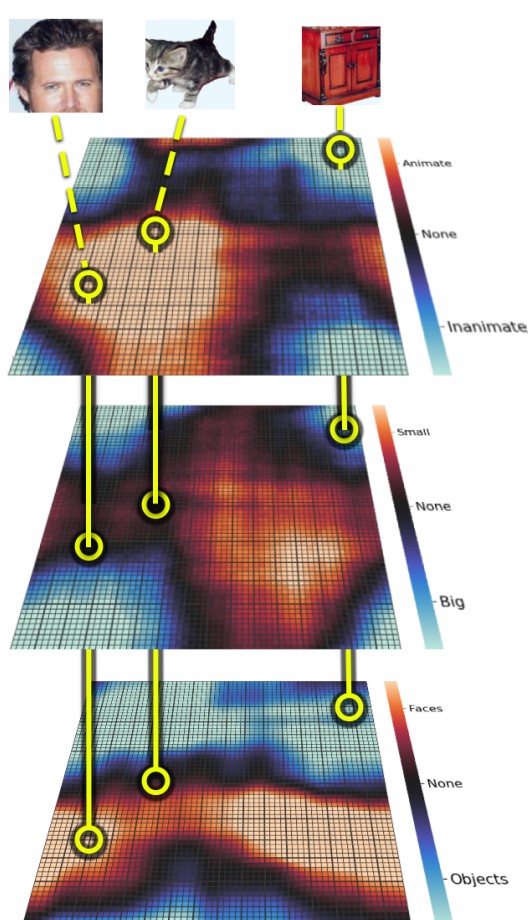

Figure 4: Selectivity maps for abstract categories: Animate vs. Inanimate (top), Small vs. Big (middle), and Faces vs. Objects (bottom). We highlight the maximum activating neurons for the individual images from Figure 3 across all maps, demonstrating their place in the proposed nested spatial hierarchy.

# Acknowledgments and Disclosure of Funding

We would like to thank the reviewers for providing helpful constructive feedback, and the organizers of the workshop for their service. Additionally, we thank the creators of Weight & Biases [9] and PyTorch [47]. Finally, we thank the Bosch Center for Artificial Intelligence for funding.

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

# A    Additional Results

## A.1    Robustness to Initialization

To verify the robustness of our results to randomness between trials, in Figure 5 below we compare the selectivity maps shown in the main text across four independant random initalizations of the weights. We first note that the emergent feature hierarchy depicted in Figure 4 appears roughly consistent across each trial. Specifically, selectivity to places, 'big', and 'inanimate' objects appears highly overlapping in each setting. We further note that the relative placement and size of the category-selective clusters (shown in the bottom row) is again roughly consistent across runs, with face and body clusters always adjacent and frequently overlapping. We see that in some runs, a small cluster selective to a generic 'object' category can be observed. The relative weakness of this cluster is likely due to the lack of uniquely identifying features shared across all images in the object dataset.

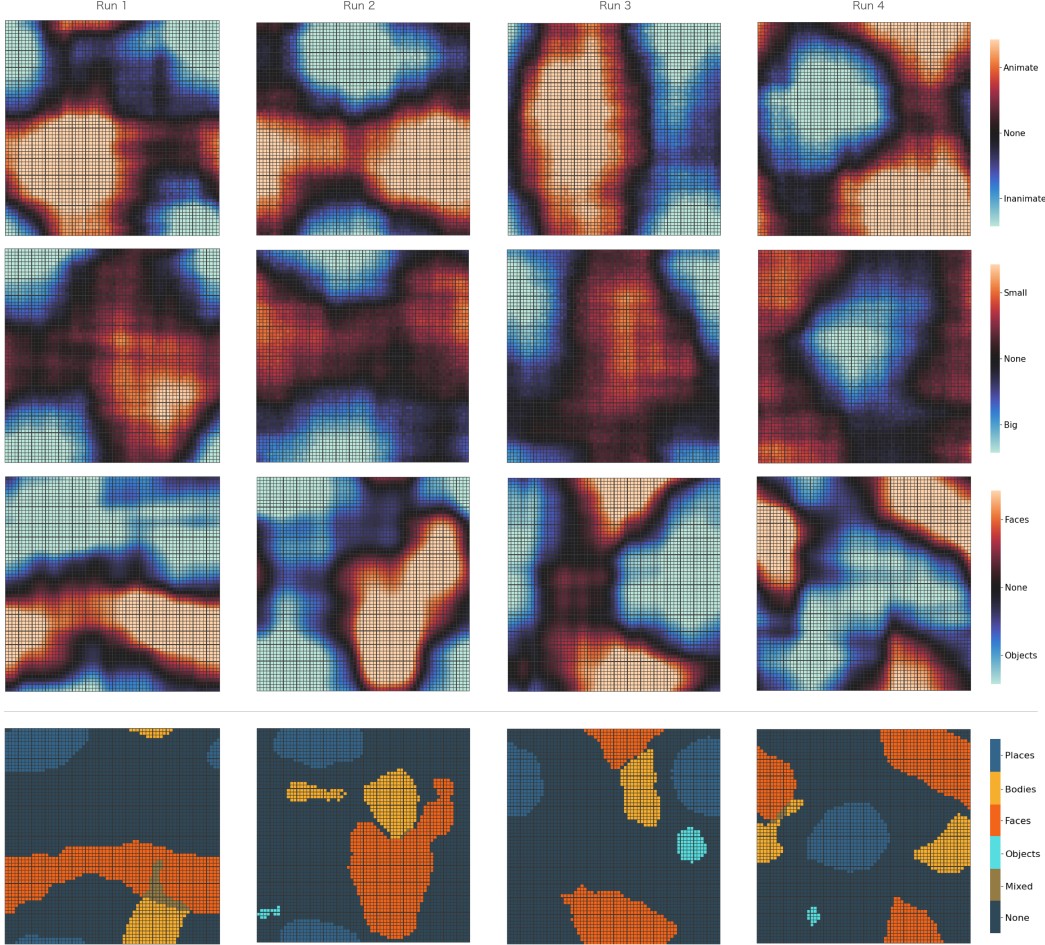

Figure 5: Selectivity maps for the TVAE across four random initalizations. We observe that the emergent feature hierarchy and the relative placement of category-clusters is consistent in each case.

## A.2    Robustness to Face Test-Dataset Choice

To investigate the robustness of face selectivity across different face test-datasets, and ensure the observed clusters are not a dataset dependant phenomenon, selectivity maps computed using four different face test-datasets are shown for both the TVAE and TDANN in Figure 6 below. Explicitly,

the four datasets included: a 25,000 subset of VGGface2 [11], 10,137 images from UTKface [64], 24,684 images from CelebA [40], and the Labled Faces in the Wild [23] dataset upon which the models were trained. The resulting selectivity maps can be seen to be highly consistent despite the variability between low-level dataset statistics, indicating the observed selectivity is more likely related to the high level category information as desired.

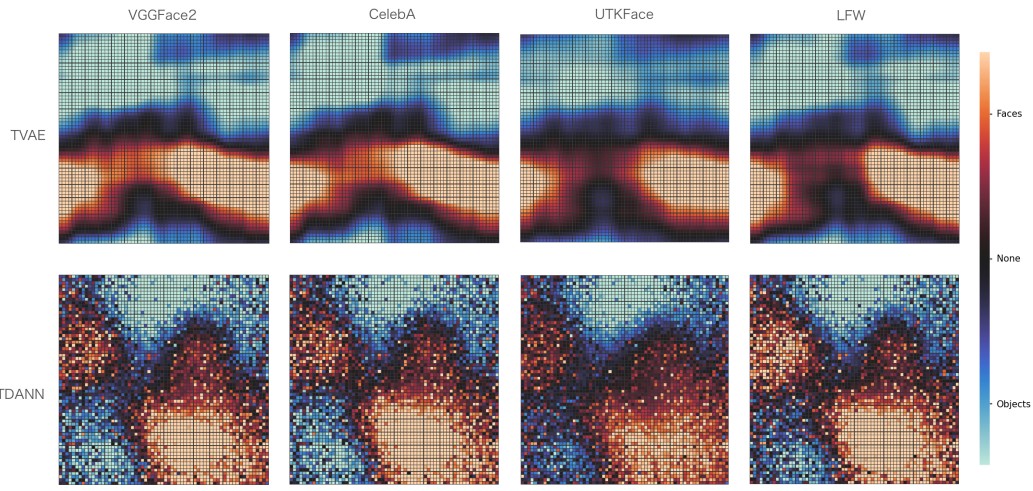

Figure 6: Face vs. Object selectivity maps for four different face datasets. We see that for both the TVAE and TDANN the relative locations and sizes of the face and object selective clusters are stable despite the differences in the underlying test datasets used.

## A.3 Distance-dependant Pairwise Correlation

To further quantify the topographic organization of the TVAE and how it compares with that of the TDANN, we measure the pairwise correlation (Pearson's R) of all topographic neurons as a function of distance in Figure 7. We see that the TDANN (right) curve matches the original results [39], roughly achieving the minimal spatial correlation loss, and mimicking the observed correlation curve from recordings in monkeys, as designed (see [39] for further discussion). Interestingly, the TVAE (middle) yields a qualitatively similar curve, despite having no such goal in its initial design. Finally, the correlation of the baseline model (left) is independant of distance as expected. We note that due to the circular boundary conditions of the TVAE, the maximal distance between neurons is significantly less, and thus scale of the X-axis is different between these two plots. In future work a more detailed comparison would benefit from matching boundary conditions in both models. Finally, in Figure 8 we plot the correlation curves for TVAEs trained with different spatial window sizes. We see that this has a significant effect on the shape of the curve, potentially allowing for more precise tuning to match biological data.

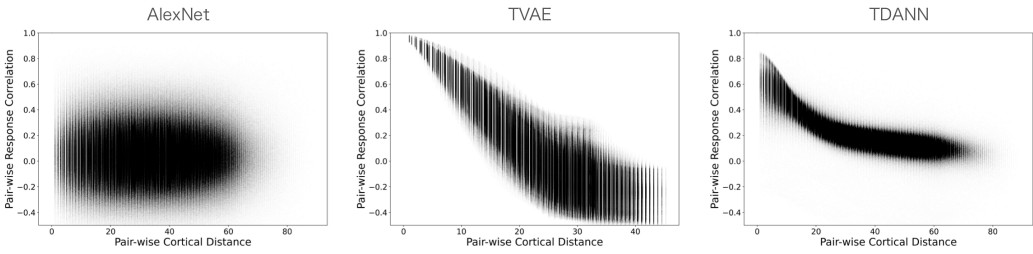

Figure 7: Pairwise correlation between neurons as a function of distance in the cortical sheet.

## A.4 Impact of TVAE Spatial Window Size (W)

In Figure 8 below, we demonstrate the effect of different choices of topographic organization (defined by $\mathbf{W}$) on the resulting learned selectivity maps. Specifically, we keep the global topography the same (a 2-d grid with circular boundary conditions), but we change the spatial extent over which variance is shared between variables $\mathbf{t}$. From left to right, we defined the matrix $\mathbf{W}$ to be a convolution matrix with kernels of size $5 \times 5$, $15 \times 15$, $25 \times 25$, and $35 \times 35$, where the total grid size is $64 \times 64$.

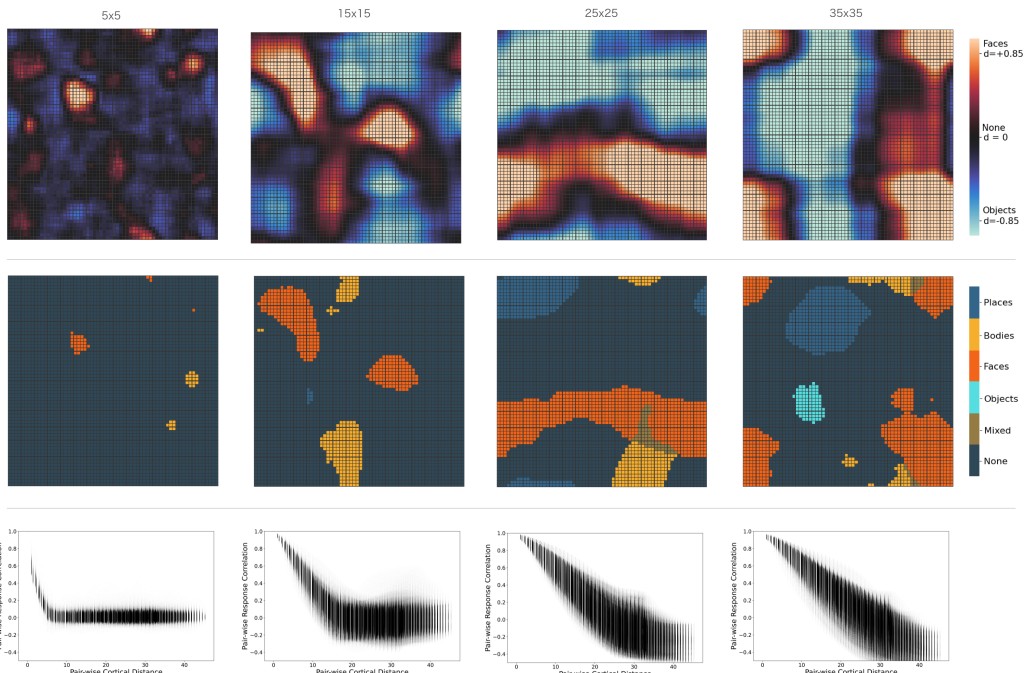

Figure 8: Selectivity maps and pairwise correlation curves for different choices of spatial window size in the Topographic VAE.

## A.5 TDANN Nested Spatial Hierarchy

In Figure 9 below, we show the abstract selectivity maps for the TDANN, analogous to those in Figure 4 for the TVAE in the main paper. We see that the TDANN does appear to have a similar nested spatial hierarchy, however it is difficult to measure the differences visually. In future work, we hope to explore methods for quantifying the extent of semantically coherent selectivity hierarchies, allowing greater comparison of models on this front.

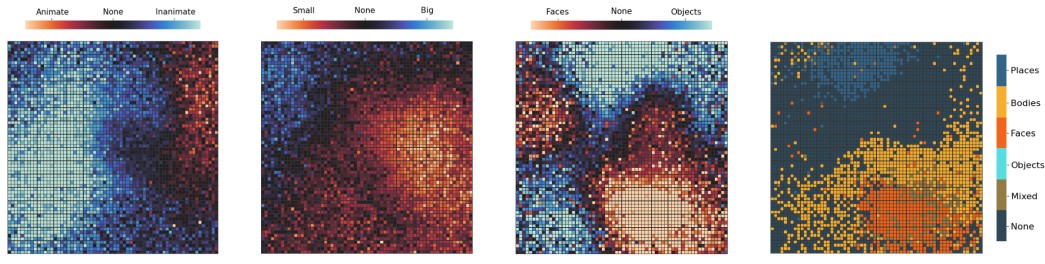

Figure 9: Abstract category selectivity for the TDANN, analogous to the results presented in Figure 4 for the TVAE. From left to right: Animate vs. Inanimate, Small vs. Big, Faces vs. Objects, and Multi-class selectivity with $d \geq 0.85$ (analogous to Figure 2).

## A.6  VAE Baseline

As an additional non-topographic baseline, we train a standard VAE in-place of the TVAE and measure the selectivity and single-image activation maps as in Figures 1 and 3. Interestingly, we see that the standard VAE exhibits significantly fewer class-selective neurons, with the majority of neurons activating for each image. We find this correlates with the measured likelihood of the data under each model, suggesting that topographic organization (and similarly class-selectivity) acts as regularization on model performance, slightly reducing the overall likelihood. As measured in prior work [38], high class-selectivity is similarly seen to be slightly detrimental to classification performance, agreeing with these results.

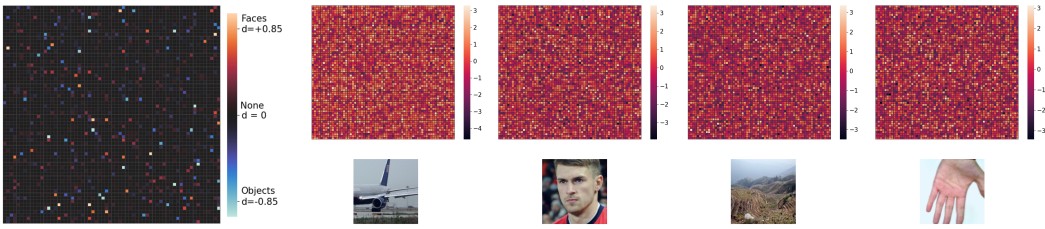

Figure 10: Face vs. Object selecitivity (left) and single-image activation maps (right) for a non-topographic VAE baseline

# B  Experimental details

All code for running the experiments in this paper can be found at the following repository:
`https://github.com/akandykeller/CategorySelectiveTVAE`

## B.1  Training details

**Dataset Preprocessing**  In order to eliminate variability between different datasets, all images were first reshaped to $256 \times 256$. A random percentage of the image area (between $8\%$ to $100\%$) and a random aspect ratio (between $\frac{3}{4}$ and $\frac{4}{3}$) were then chosen, and each image was then cropped according to these values. Finally, the crops were resized to the final shape of $224 \times 224$. All images were then normalized by the mean $[0.48300076, 0.45126104, 0.3998704]$ and standard deviation $[0.26990137, 0.26078254, 0.27288908]$.

**TDANN Hyperparameters**  The TDANN model was trained with stochastic gradient descent, a learning rate of $1 \times 10^{-3}$, standard momentum of 0.9, and a batch size of 128 for 10 epochs. Explicitly, the loss function was given by a sum of the classification cross entropy loss, the spatial correlation losses for both layers FC6 and FC7, and weight decay of $5 \times 10^{-4}$. A fixed weight of $10 \times \frac{1}{4096^2}$ was multiplied by the spatial correlation loss before backpropagating as this was found necessary to qualitatively match the results from Lee et al. [39]. Contrary to the original TDANN work, we did not randomly initialize the locations of the neurons, and instead spaced them evenly on a grid of the same size. We found the spatial correlation loss to still function equally well in this setting, and detail our implementation in Section B.2 below.

**TVAE Hyperparameters**  The TVAE was trained with stochastic gradient descent, a learning rate of $1 \times 10^{-5}$, standard momentum of 0.9, and a batch size of 128 for 30 epochs. The global topology was set to a single 2D torus (i.e. a 2D grid with circular boundary conditions), and the local topology was set to sum of local regions of size $25 \times 25$, i.e. the kernel used to convolve over **u** was of size $25 \times 25$ and contained all 1's. The $\mu$ parameter was initialized to 40, and trained simultanously with the remainder of the model parameters.

## B.2 Spatial Correlation Loss of TDANN

The exact form of the spatial correlation loss used for training the TDANN in this paper is given as:

$$\text{SpatialCorrelationLoss}(\mathbf{z}) = \sum_i^n \sum_{j \neq i}^n \left| C_{ij}(\mathbf{z}) - \frac{1}{D_{ij} + 1} \right| \tag{5}$$

where $\mathbf{z}$ an $n$-dimensional vector of activations, $C$ is the normalized cross correlation matrix (e.g. a matrix of Pearson correlation coefficients), and D is a matrix containing the 'cortical distances' in millimeters between all pairs of neurons $i$ and $j$. In this work, we defined all neurons to be equally spaced in a 2-D grid of 10mm × 10mm. This resulted in a horizontal and vertical spacing between neurons of 0.15625mm and a diagonal spacing of 0.22097087mm. Unlike the TVAE, the TDANN grid was not defined to have circular boundary conditions in order to match the original model.

