# OpenReview forum: "Modeling Category-Selective Cortical Regions with Topographic Variational Autoencoders"
_NeurIPS.cc/2021/Workshop/SVRHM — SVRHM 2021 Oral_

### Official Review · Reviewer_3GW3 · 2021-10-16
**Well-motivated modelling of category-selective regions using TVAE**

**Rating:** 8
**Confidence:** 4

**Review:**

The authors used the topographic variational autoencoders to show that computational models training with topographical structure could yield human-like category-selective regions within the network. The theocratical contribution of this paper address the question on how anatomical constraints (i.e. wiring length) may lead to categorical selectivity observed in the human brains.

Pros:
1.	The logic of the paper is clear, and the figures are easy to read, and address the questions beautifully.
2.	It is clever for the authors to use both TDANN and TVAE to show that the categorical selectivity within network exists in both models with supervised and non-supervised training.
3.	The “nested spatial hierarchy of categories” analysis is very interesting, and could be extended to see how the selectivity to different abstract properties (e.g. animacy, real-world size) overlap with the selectivity to different object classes.

Suggestions:
1.	While the paper mentioned the TDANN results, it will be beneficial for the authors to discuss the similarities and differences between TDANN and TVAE, such that readers without in-depth knowledge on TDANN could understand it.
2.	The baseline model used in this paper is the AlexNet model, which performs classification task. However, TVAE is an auto-encoder, so it would be nice to have another auto-encoder model without topographical constraints in training as the baseline model to demonstrate that the topographical constraint is essential for the categorical selectivity.
3.	As the authors discussed, it is important to understand whether the current results emerge from the class-level features from the pre-trained AlexNet features, so a full end-to-end training could be beneficial in future work.

---

> ### Author Response · Authors · 2021-12-10
> **Response to Reviewer 3GW3**
>
> We thank Reviewer 3GW3 for their time spent with our paper and their helpful suggestions. We have included their proposed additional baseline in the appendix and agree that end-to-end training would be of the utmost interest. Finally, we have aimed to define the TDANN more precisely in the related work section, allowing for a better understanding of how it relates to the TVAE.

---

### Official Review · Reviewer_yFqU · 2021-10-29
**Modeling Category-Selective Cortical Regions with Topographic Variational Autoencoders**

**Rating:** 9
**Confidence:** 4

**Review:**

This is a very innovative and interesting paper. The authors demonstrate that category selectivity, akin to that observed in primate visual cortex, arises naturally in topographic variational autoencoder models. These models not only show face/object selectivity, but also more broad organization for real-world size and animacy observed in the human brain. I think this is a really exciting result with many interesting future applications for cognitive neuroscientists. I do have several suggestions to improve the paper, listed below:

The term TDANN is never introduced, and the acronym is never defined! While these may be famliar to several in the field, they are still quite new and given the interdisciplinary audience at SVRHM require some introduction. The acronym shoudld be spelled out in the abstract/intro and  This should be spelled out in the abstract and they should be briefly described in the background section.
I found the brief commentary of divisive normalization (lines 63-64) and the background section on topographic ICA somewhat confusing. It also wasn’t clear that they provided much intuitive understanding. I would suggest the note on divisive normalization either be removed or reworded and the section on topographic ICA be shortened to focus only on the most relevant portions (if any) for understanding
It may help to provide a more intuitive introduction to standard VAEs before jumping into the description of topographic VAEs
Several parts of the paper referred to research in two papers referenced only by the first author (Lee et al and Zhang et al). This made it difficult to follow since the reader has to flip back and forth to the references. I would suggest introducing the papers (incl first authors) in a sentence up front and then referring to them by first author et al, or (better) providing a more intuitive description for each paper/work.
If there is space, a brief model schematic might help clarify the section on Models in 129.
Figure 1: What is the range on this color bar? Were any significance tests performed?
Figure 3: How does this compare to a TDANN?
Another useful baseline might be a standard VAE without any topographic constraints
The results presented in figure 4 are very cool!
If there is space in the discussion, it would help to provide some thoughts on what implications such a “redundancy reduction” has for the human brain.

---

> ### Author Response · Authors · 2021-12-10
> **Response to Reviewer yFqU**
>
> We thank Reviewer yFqU for their interest and excitement in our paper. Additionally, we greatly appreciate the noted issues with clarity which we hope they find improved upon in the main paper. Finally, we have included their proposed updates to the figures and additional baselines in the appendix.

---

### Official Review · Reviewer_TJyg · 2021-10-30
**A computational reason for topographic organization in the ventral stream**

**Rating:** 8
**Confidence:** 4

**Review:**

"Modeling Category-Selective Cortical Regions with Topographic Variational Autoencoders" attempts to understand the topography of the ventral visual stream using a new class of topographic model recently introduced by Keller and Welling (2021): topographic variational autoencoders. This is a welcome addition to a growing literature modeling the topographic organization of the visual system, coming at the problem from a generative modeling perspective. The authors perform a neat set of experiments to show that their model develops topographic organization similar to the ventral stream, and to compare their model to another recently developed model of topographic organization, the Topographic Deep Artificial Neural Network. The paper is very well written and easy to follow. Overall, this will be an interesting contribution to the workshop.

Pros:
1) the paper provides a good literature survey of both the problem (topographic organization in the ventral stream) and algorithms that have been developed to model the problem, including a strong background specifically on topographic generative models, of which the TVAE is an instance.
2) the paper makes an interesting connection between the local correlation wiring length minimization proxy of the TDANN and the statistical basis for topographic generative models. the authors might consider adding a sentence to also connect their work with the Interactive Topographic Network (ITN; Blauch, Behrmann, Plaut, 2021) which induces local correlation through locally-biased excitatory feedforward connections.
3) the TVAE demonstrates very similar selectivity to the ventral stream, whereby both category-selectivity (e.g. face selectivity) and attribute-selectivity (e.g. animacy, size selectivity) are seen and overlap in what the authors refer to as a nested spatial hierarchy. For example, the cluster of face-selective units largely appears in the animate-preferring zone.


Cons:
1) the authors claim that their model is fully unsupervised but it is learned on top of pre-trained features acquired from a supervised Alexnet. moreover, the TVAE has not yet been shown to perform well on large-scale tasks, as the recent paper which introduced it worked with MNIST. can the authors comment on whether an identical architectural setup would be capable of learning representations end to end, even if stacked on top of a (untrained) convolutional feature extractor? relatedly, the authors claim that the limitation of working with pre-trained features is shared by all previous works modeling topographic organization of VTC, but this is false. The TDANN was trained end to end in Lee et. al (2020), and some instances of the ITN were trained end-to-end in Blauch et. al (2021) -- however, these models do all share the limitation of working with convolutional feature extractors.
2) moreover, the ability of the TVAE to support downstream tasks is not discussed at all (performance is only quantified in terms of the model's generative capacity). Can the authors perform a linear readout from the TVAE onto the imagenet categories and compare with the TDANN?

Possible points for improvement:
1) the authors claim that there are two potential explanations of the origins of topographic organization -- anatomical constraints, in which they include the theory of wiring minimization and cite the TDANN and SOM, and information theory, in which they cite learned specialization in DNNs. However, these explanations are really complementary rather than competitive. The reason that wiring minimization should give rise to topographic organization is because there is specialization inherent to the neural code that can be mapped anatomically. So perhaps the wiring minimization framework is really the link between anatomical constraints (perhaps, innate pre-specification of specialization) and information theoretical accounts.
2) the authors mention the possibility of a hierarchical extension of the TVAE. i am curious why this has not already been explored -- a matched TVAE to the TDANN model tested would be hierarchical. so, can the authors comment on whether this is particularly difficult in the TVAE?
3) the authors should clarify which layer of alexnet served as the input to the TVAE.
4) it would be interesting to see the distance-dependent pairwise correlation relationship of the TVAE, especially compared with the TDANN which was trained to generate a certain relationship.
5) the analysis of the nested spatial hierarchy is very preliminary and only visual. the authors might consider doing a more in depth analysis of the relationship between different levels of the hierarchy. also, is this same pattern seen in the TDANN?
6) can the authors comment on the effect of the spatial window size of the TVAE on the learned topography and possibly performance?
7) compared to the TVAE in Keller & Welling (2021), it appears the TVAE used here does not have a disjoint topological (i.e. capsule) structure, and does not utilize temporal coherence. it would be interesting for the authors to comment on these choices.

---

> ### Author Response · Authors · 2021-12-10
> **Response to Reviewer TJyg**
>
> We thank Reviewer TJyg for their thorough review and helpful comments. As mentioned in our main response, we have attempted to ameliorate the inaccuracies pointed out by the reviewer, and thank them for bringing these to our attention. Additionally, we have attempted to answer the majority of their remaining proposed points for improvement in the appendix. To answer the remaining questions:
> - We have not explored hierarchical extensions to the TVAE simply due to time constraints. We believe the extension should be relatively straight-forward, with the main area of uncertainty relating to the choice of ordering of the latent variables for the inference and generative models which may take some experimentation to resolve optimally.
> - The TVAE in this paper can be seen as having a single 2-D capsule instead of the multiple disjoint 1-D capsule structures demonstrated in the original TVAE paper. This is mainly due to the dataset structure used in this paper being individual images rather than transformation sequences. Figure 3 of the original TVAE paper actually shows a single 2-D capsule model trained on MNIST equivalent to the one used in this paper -- and MNIST-class-selective clusters can additionally be observed.

---

### Decision · Program_Chairs · 2021-11-02

Accept (Oral)

---

> ### Author Response · Authors · 2021-12-10
> **Author Response: Camera-Ready Changes**
>
> We thank the reviewers for their constructive feedback and the significant time they spent reviewing our work.
>
> To address the reviews we have updated the camera-ready version of the paper with the following additions:
> - We have updated the introduction to reflect Reviewer TJyg's comment regarding the potential complimentary, rather than competitive nature of the proposed explanations for topographic organization.
> - We have included a new Related Work section to specifically discuss and define the TDANN, VTC-SOM, and ITN models, hopefully improving clarity of the paper. Additionally, we have corrected the inaccuracies with respect to our description of prior work as mentioned by Reviewer TJyg.
> - We have included a breif background section on VAEs to provide a more complete overview for the interdisciplinary audience at SVRHM, as pointed out by Reviewer yFqU.
> - We included a section discussing the impact of topographic organization on model performance to validate the downstream performance of the proposed models, and validate that the learned representations are not degenerate.
> - We have included ranges for color bars on the selectivity maps, and clarified the details of which layers are being displayed, as well as which AlexNet layers are used for feature extraction.
>
> In the appendix, we have included the following additional results:
> - We show selectivity maps from four random initializations of the TVAE to verify robustness and significance of the provided results.
> - We provide the distance-dependant pairwise correlation curves for all models presented, as suggested by Reviewer TJyg, allowing for closer comparison with the original TDANN work.
> - We have provided an initial study of the impact of the spatial window size of the TVAE on the demonstrated selectivity maps and correlation curves.
> - We included selectivity and activation maps for a baseline non-topographic VAE model, as suggested by Reviewers yFqU and 3GW3.
> - We provide the abstract selectivity maps for the TDANN, allowing for comparison with the TVAE on this front.